# Reticulocalbin 3 Is a Novel Mediator of Glioblastoma Progression

**DOI:** 10.3390/cancers15072008

**Published:** 2023-03-28

**Authors:** Yi He, Salvador Alejo, Jessica D. Johnson, Sridharan Jayamohan, Gangadhara R. Sareddy

**Affiliations:** 1Department of Obstetrics and Gynecology, University of Texas Health San Antonio, San Antonio, TX 78229, USAalejo@livemail.uthscsa.edu (S.A.);; 2Department of Neurosurgery, Xiangya Hospital, Central South University, Changsha 410008, China; 3Mays Cancer Center, University of Texas Health San Antonio, San Antonio, TX 78229, USA

**Keywords:** reticulocalbin 3, glioblastoma, glioma stem cells, ribosome, endoplasmic reticulum, translation

## Abstract

**Simple Summary:**

Glioblastoma is the deadliest primary brain tumor. Current treatment strategies for glioblastoma are not effective, and patients exhibit poor survival rates. Emerging studies implicated glioma stem cells (GSCs) in tumor initiation, progression, and therapy resistance. There is an unmet need to identify new therapeutic candidates for glioblastoma. Our study showed that reticulocalbin 3 (RCN3), an ER lumen residing Ca^2+^ binding protein, is overexpressed in glioblastoma and associated with poor survival rates. The reduction in RCN3 expression using shRNA or gRNA resulted in reduced proliferation and self-renewal of GSCs. The RNA-seq results showed that RCN3 knockdown altered the expression of several genes related to translation, ribosome, stem cell differentiation, and extracellular matrix. In silico analysis of glioblastoma patient datasets demonstrated a positive correlation of RCN3 with ribosomal pathway genes. Importantly knockdown of RCN3 significantly enhanced the survival of tumor-bearing mice. Our study suggests that RCN3 could be a potential therapeutic target in glioblastoma.

**Abstract:**

Glioblastoma is the most common malignant primary brain tumor. Molecular mechanisms underlying the pathobiology of glioblastoma are incompletely understood, emphasizing an unmet need for the identification of new therapeutic candidates. Reticulocalbin 3 (RCN3), an ER lumen-residing Ca^2+^ binding protein, plays an essential role in protein biosynthesis processes via the secretory pathway. Emerging studies demonstrated that RCN3 is a target for therapeutic intervention in various diseases. However, a knowledge gap exists about whether RCN3 plays a role in glioblastoma. Publicly available datasets suggest RCN3 is overexpressed in glioblastoma and portends poor survival rates. The knockdown or knockout of RCN3 using shRNA or CRISPR/Cas9 gRNA, respectively, significantly reduced proliferation, neurosphere formation, and self-renewal of GSCs. The RNA-seq studies showed downregulation of genes related to translation, ribosome, and cytokine signaling and upregulation of genes related to immune response, stem cell differentiation, and extracellular matrix (ECM) in RCN3 knockdown cells. Mechanistic studies using qRT-PCR showed decreased expression of ribosomal and increased expression of ER stress genes. Further, in silico analysis of glioblastoma patient datasets showed RCN3 expression correlated with the ribosome, ECM, and immune response pathway genes. Importantly, the knockdown of RCN3 using shRNA significantly enhanced the survival of tumor-bearing mice in orthotopic glioblastoma models. Our study suggests that RCN3 could be a potential target for the development of a therapeutic intervention in glioblastoma.

## 1. Introduction

Glioblastoma (Glioblastoma, IDH-wildtype) [1] accounts for 50.1% of all primary malignant brain tumors, and 14,490 new cases are projected in 2023 with an annual incidence of 3.26 per 100,000 population in the United States [2]. Despite decades of unrelenting efforts made to cure this disease, the median survival of glioblastoma patients is approximately 8 months, and only 6.9% of patients survived five years post-diagnosis [2]. The current standard of care for glioblastoma consists of surgical debulking of the tumor followed by concurrent chemoradiation therapy using the alkylating agent temozolomide [3]. Unfortunately, most tumors will recur due to chemo- and radiotherapy resistance, and patients will succumb to it eventually. Glioblastoma pathogenesis is driven by alterations in tumor suppressors as well as oncogenes [4,5]. Several distinct features of glioblastoma attributed to the therapy resistance, including highly infiltrative and heterogeneous nature [6,7], complex tumor microenvironment [8], low immunogenicity [9], cancer stem-like features [10], blood-brain barrier, and chemotherapy resistance mechanisms [11]. Emerging studies implicated glioma stem cells (GSCs) in tumor initiation, progression, and therapy resistance of glioblastoma [10,12,13]. Specific mechanisms of glioblastoma pathogenesis are still unclear, and the efficacy of current target molecules is unsatisfactory, making it an unmet need to identify novel targets for glioblastoma treatment.

Reticulocalbin 3 (RCN3) belongs to the CREC (Ca^2+^ binding protein of 45 kDa (Cab45), reticulocalbin, ER Ca^2+^ binding protein of 55 kDa (ERC-55), and calumenin) family of multiple EF-hand Ca^2+^ binding proteins localized to the secretory pathway [14]. CREC family consists of five members RCN1, RCN2, RCN3, SDF4, and CALU [15]. Recent studies showed that these proteins are not solely localized to the secretory pathway but are also found in the cytosolic compartment and at the cell surface. RCN3 gene is located on chromosome 19q13.33 and encodes a Ca^2+^-binding protein containing EF-hands that can assist protein biosynthesis and transportation in the endoplasmic reticulum [15,16,17]. Structurally, RCN3 consists of a signal sequence, six EF-hands, and an HDEL ER-retention signal [15]. Recent studies showed that RCN3 contains five R-X-X-R motifs, which represent the target sequence of subtilisin-like proprotein convertases [15,17]. In addition to its role in the secretory pathway, RCN3 also regulates several biological processes, including primordial follicle formation [18], collagen fibrillogenesis, and tenocyte maturation during postnatal tendon development [19]. Moreover, it plays an essential role in perinatal lung maturation and neonatal respiratory adaptation [20]. Emerging studies revealed RCN3′s association with cancer. Single-cell multi-omics sequencing studies identified RCN3 as a biomarker for the poorer prognosis of colorectal cancer [21]. Further, RCN3 is overexpressed in melanoma [22] and esophageal squamous cell carcinoma [23], whereas it is downregulated in NSCLC [24] and osteosarcoma [25]. However, it is currently unknown whether RCN3 plays a role in glioblastoma progression.

In this study, we investigated the role of RCN3 in glioblastoma pathogenesis. Using public datasets, we identified that RCN3 is upregulated in glioblastoma and associated with poor overall survival. Knockdown or knockout of RCN3 decreased the proliferation and self-renewal ability of GSCs. RNA-seq and mechanistic studies revealed that RCN3 regulates ribosomal, stem cell differentiation, and cytokine pathways. Further, the knockdown of RCN3 improved the survival of tumor-bearing mice.

## 2. Materials and Methods

### 2.1. Gene Expression Analysis

The expression levels of *RCN3* between tumor patients and normal individuals were explored with the UCSC XENA platform (https://xenabrowser.net/datapages/ accessed on 1 June 2021). mRNA expression data (TPM transformed from HTSeq-FPKM) and related clinical information about glioblastoma patients were downloaded from the TCGA database (https://cancergenome.nih.gov accessed on 1 June 2021). Samples without clinical features were excluded. In the end, a total of 172 glioblastoma patients’ data were included in this study. TPM format RNAseq data of normal brain tissue was extracted from GTEx (1152 samples) and TCGA (5 samples) datasets in UCSC XENA and analyzed with R. The expression difference in RCN3 protein between a normal brain and glioblastoma samples was explored with IHC results from the Human Protein Atlas platform (HPA, https://www.proteinatlas.org/ accessed on 1 June 2021).

### 2.2. Diagnostic and Prognostic Value of RCN3 in Glioblastoma

Receiver operating characteristic (ROC) curve was applied to evaluate the diagnostic prediction efficacy of the RCN3. Kaplan–Meier analysis was utilized to examine the overall survival (OS) difference between groups with different RCN3 expressions (lowest 25% vs. highest 25%). Thereafter, through incorporation of RCN3 expression and other clinical factors, a nomogram was constructed to predict prognosis of a specific patient.

### 2.3. Cell Culture, Reagents, and Generation RCN3 Knockdown and Knockout Cells

Human HEK293T cell line was obtained from the American Type Culture Collection (ATCC, Manassas, VA, USA) and cultured in DMEM supplemented with 10% fetal bovine serum (Sigma Chemical Co., St. Louis, MO, USA). Patient-derived primary glioma stem cells (GSCs) were established from discarded specimens obtained from glioblastoma patients undergoing surgery at UT Health San Antonio. The specimens were collected in accordance with the Declaration of Helsinki and approved by the Institutional Review Board (or Ethics Committee) of UT Health San Antonio. Patients provided informed consent for surgery and use of their tissues for research. Deidentified fresh tissue was collected intraoperatively, dispersed into single cells, and cultured briefly in a neurobasal media for expansion and purification, followed by intracranial injection in nude mice. Patient-derived GSC040815 and GSC082209 cells were cultured as neurospheres in neurobasal medium (Invitrogen, Carlsbad, CA, USA) supplemented with B27 serum-free supplement, EGF (20 ng/mL), bFGF (20 ng/mL), LIF (10 ng/mL), and heparin (5 µg/mL) as described previously [26]. Human neural stem cell line (cat # hNSC11) derived from iPSCs of neonatal foreskin fibroblasts was purchased from ALSTEM, Inc. (Richmond, CA, USA). RCN3 antibody (Cat # ab204178) was purchased from Abcam (Cambridge, MA, USA). GAPDH (Cat # 8884) and CD44 (Cat # 37259S) antibodies were obtained from Cell Signaling Technology (Beverly, MA, USA). RCN3-specific shRNA (Cat # TRCN0000029495; target sequence: GACAGAAACAAAGATGGCTAT) and nontargeting control shRNA (Cat # SHC016-1EA) lentiviral plasmids were purchased from MilliporeSigma (Burlington, MA, USA). Scrambled gRNA (pLV[CRISPR]-hCas9:T2A:Bsd-U6>Scramble[gRNA#1]) and RCN3-specific gRNA1 (pLV[CRISPR]-hCas9:T2A:Bsd-U6>hRCN3[gRNA#1272]; guide sequence-ACGGACCGCGACGGGCGTGT) and gRNA2 (pLV[CRISPR]-hCas9:T2A:Bsd-U6>hRCN3[gRNA#1279]-guide sequence-ACCTATGGCCACTACGCGCC) lentiviral plasmids were purchased from Vector Builder. shRNA- or gRNA-specific lentiviral particles were generated by transfecting pMDLg/pRRE (Cat # 12251), pMD2.G (Cat # 12259), and pRSV-Rev (Cat # 12253) packaging plasmids (purchased from Addgene) together with transfer plasmid into HEK293T cells using TurboFect Transfection reagent (Thermo Scientific, Waltham, MA, USA). Supernatant media was collected at 48 h and 72 h post-transfection, and lentivirus was concentrated using Lenti-X Concentrator (Takara Bio, San Jose, CA, USA). GSCs stably expressing shRNA or gRNA were generated by infecting cells with human-specific RCN3-shRNA or gRNA lentiviral particles, and stable clones were selected with puromycin (1 μg/mL) or blasticidin (5 μg/mL), respectively. Lentiviral particles expressing nontargeting control shRNA or scramble gRNA were used to generate control cells. To prevent the artifact of choosing a single clone, we have employed pooled clones.

### 2.4. Cell Viability, Neurosphere Formation, and Extreme Limiting Dilution Assays

The effect of RCN3 knockdown or knockout on the cell viability of GSCs was determined using CellTiter-Glo 2.0 Cell Viability Assay (Promega Cat# G9241, Madison, WI, USA) according to the manufacturer’s instructions. GSCs that stably express either control shRNA or RCN3 shRNA and alternatively scramble or one of two RCN3-specific gRNA were seeded in 96-well, flat, clear-bottom, opaque-wall microplates. The total ATP content as an estimate of the total number of viable cells was measured using Promega™ GloMax^®^ Luminometer. For neurosphere assays, single-cell suspensions of GSCs were seeded in 24-well plates (20 cells per well) in quadruplicate and cultured for 10 days. Then, newly formed neurospheres were imaged and measured using the NIH ImageJ software v1.53 and Pombe Measurer plugin v1.0. Sphere diameter values were plotted on a histogram. Extreme limiting dilution assay (ELDA) was performed by seeding decreasing numbers (50, 20, 10, 5, and 1 cells/well) of indicated GSCs in 96 well ultra-low attachment plates. Fourteen days later, the number of wells containing spheres was counted and used to calculate the frequency of self-renewing GSCs by the ELDA software (http://bioinf.wehi.edu.au/software/elda/ accessed on 15 January 2023) and statmod package in R.

### 2.5. RNA-seq and RT-qPCR

RNA-seq experiments were performed as described previously [27]. RNA was isolated from control and RCN3 knockdown GSCs (GSC 040815) using RNeasy mini kit (Qiagen Cat# 74104). RNA was subjected to RNA-seq on Illumina HiSeq 3000 system using Illumina TruSeq stranded mRNA-seq library preparation kit by UTHSA Genome Sequencing Facility. The sequenced results were deposited in the GEO database under a GEO accession number (GSE226990). Differentially expressed genes (DEGs) were determined using DESeq2 package in R. Protein-protein-interaction (PPI) analysis of DEGs was performed in STRING platform (https://cn.string-db.org/cgi/input.pl accessed on 20 November 2022) and further enriched in Cytoscape software with MCODE application (https://cytoscape.org/ accessed on 1 June 2021) to obtain the key module genes. Biological significance of the genes was determined using Gene set enrichment analysis (GSEA). For each analysis, gene set permutation was performed 1000 times. Gene sets with a nominal *p*-value below 0.05 and false discovery rate (FDR) below 0.25 were considered as significantly enriched. To support the RNAseq results, Gene Ontology (GO) and Kyoto encyclopedia of genes and genomes (KEGG) functional enrichment analysis of DEGs between high and low RCN3 expression samples were also performed with glioblastoma data in mRNA-693 dataset from CGGA (Chinese Glioma Genome Atlas, http://www.cgga.org.cn/index.jsp accessed on 1 June 2021). RNA expression file of glioblastoma samples was acquired from CGGA-mRNA-693 dataset and then divided into high and low RCN3 expression groups based on RCN3 expression level (median as cutoff value). The DEGs between groups were identified using “limma” package with |log2FC| > 1 and padj < 0.01 as significance threshold and used for functional enrichment of GO and KEGG pathways. R packages used for analysis and visualization include “limma”, “org.Hs.eg.db”, “DOSE”, “clusterProfiler”, “enrichplot”, “scatterplot3d”, “ggplot2”, “circlize”, “ggpubr”, “colorspace”, “stringi”, and “RColorBrewer”. RT-qPCR was performed by synthesizing cDNA from RNA using Applied Biosystem cDNA synthesis kit, and real-time PCR was carried out using SYBR green master mix and gene-specific primers. The primer sequences of genes were obtained from Harvard Primer Bank (http://pga.mgh.harvard.edu/primerbank/ accessed on 7 November 2022), and sequences are shown in Appendix A.

### 2.6. Western Blotting

Whole-cell lysates were prepared from GSCs using RIPA buffer containing protease and phosphatase inhibitors (Sigma Chemical). Total proteins (20 μg) were mixed with SDS sample buffer and subjected to SDS–PAGE, and the resolved proteins were transferred onto nitrocellulose membranes. Blots were then blocked with 5% nonfat dry milk powder for 1 h at room temperature and incubated with primary antibodies (RCN3-1:1000; GAPDH-1:1000) overnight at 4 °C followed by incubation with secondary antibodies for 1 h (anti-rabbit IgG; Cytiva NA9341ML- 1:2000 dilution) at room temperature. Blots were developed using the ECL kit (Thermo Scientific, Waltham, MA, USA) and imaged with BioRad Chemidoc system. The uncropped blots and molecular weight markers are shown in Appendix A.

### 2.7. In Vivo Orthotopic Tumor Model

All animal experiments were performed after obtaining IACUC approval and by following UTHSA institutional guidelines. NOD.CB17-Prkdc^scid^/NCrCrl male mice of 8–10 weeks old were purchased from Charles River (Wilmington, MO, USA). Control and RCN3 knockdown GSCs (5 × 10^4^) were injected orthotopically into the mice’s right cerebrum. Throughout the study, animals were monitored for neurological symptoms. Mice were euthanized, and brains were isolated and processed for histological studies. Mouse survival was determined using Kaplan–Meier survival curves and log-rank test using GraphPad Prism 9 software v9.5 (GraphPad Software, San Diego, CA, USA).

### 2.8. Immunohistochemistry

Immunohistochemistry (IHC) experiments were performed as described [27]. Briefly, tumor sections were incubated with RCN3 antibody (Abcam Cat# ab204178—1:500 dilution), Ki67 antibody (Abcam Cat# ab16667—1:100 dilution), or CD44 antibody (Cell Signaling Cat# 37259S—1:200 dilution) overnight, followed by secondary antibody incubation for 30 min (Vector Labs Cat# MP-7500). Immunoreactivity was detected using DAB staining, and sections were counterstained with hematoxylin.

### 2.9. Statistical Analyses

The expression level of the *RCN3* gene in glioblastoma patients was evaluated with box plots. Median was used as cut-off value for high and low *RCN3* expression. Wilcoxon test was employed for differential analysis, and Spearman method was applied for correlation analysis. All hypothesis tests were two-sided, with * *p*  <  0.05, ** *p*  <  0.01, *** *p* < 0.001, and **** *p* < 0.0001. All other statistical analyses and data visualizations were carried out in R software (R version 4.1.2).

## 3. Results

### 3.1. RCN3 Is an Upregulated Prognostic Marker of Poor Survival in Glioblastoma

We first examined the expression difference in RCN3 in a pan-cancer scope using the UCSC XENA database (Figure 1A). RCN3 is significantly overexpressed in multiple tumors, such as glioblastoma, gastric cancer, sarcoma, and esophageal cancer. Next, we compared its expression levels between normal and glioblastoma tissues and found that RCN3 gene and protein levels are significantly higher in glioblastoma compared to normal brain tissues (Figure 1B–E). Further glioblastoma molecular subgroup expression analysis showed that RCN3 is highly expressed in the mesenchymal subtype in TCGA and CGGA data sets (Figure 1F; Appendix A). Further exploration of TCGA and CGGA data sets showed significantly higher RCN3 expression in IDH wild-type tumors compared to IDH mutant tumors (Figure 1G; Appendix A). Importantly, the Kaplan–Meier survival analysis showed that higher RCN3 expression in glioblastoma patients was associated with poorer overall survival (Figure 1H). This was consistent with ROC (receiver operating characteristic) curve analysis showing that RCN3 expression is strongly linked with a high diagnostic value for glioblastoma with an AUC (area under curve) of 0.901 (Figure 1I). Thereafter, we integrated the clinicopathological variables and RCN3 expression level to construct a nomogram to predict the 1-, 3-, and 5-year survival probabilities of glioblastoma patients (Figure 1J). Further prognosis correlation analysis using the TIMER2.0 database demonstrated that the RCN3 gene is associated with the prognosis of multiple tumors, including glioblastoma (Figure 1K). Altogether, these results suggest that RCN3 is overexpressed in glioblastoma and might play an essential role in glioblastoma pathobiology.

### 3.2. RCN3 Knockout or Knockdown Inhibited Proliferation, Neurosphere Formation, and Self-Renewal of GSCs

Glioma stem cells (GSCs) often contribute to tumor initiation, progression, and therapy resistance of glioblastoma [12,28,29,30,31], and their elimination is critical for the development of efficient therapeutic strategies [32]. Since RCN3 is highly expressed in glioblastoma and correlates with worse prognosis, we sought to determine if its expression is elevated in patient-derived GSC models versus human neural stem cells (hNSCs). RT-qPCR results showed that RCN3 is highly expressed in GSCs compared to hNSCs (Figure 2A). To understand the functional role of RCN3, we transduced two patient-derived GSCs with scramble or one of two different RCN3-specific gRNAs, and the knockout of RCN3 was confirmed using Western blotting (Figure 2B). Next, we examined the cell proliferation rates of GSCs using Cell-Titer Glo assays. As shown in Figure 2C,D, the knockout of RCN3 significantly reduced the proliferation of GSCs compared to controls. Further, the self-renewal ability of scramble or RCN3 gRNA transduced GSCs was determined using neurosphere formation and extreme limiting dilution assays. As shown in Figure 2E,F, RCN3 knockout significantly reduced the neurosphere growth of GSCs compared to controls. Using limiting dilution assay, we found RCN3 knockout significantly reduced the GSC self-renewal compared to controls (Figure 2G,H). Further, RCN3 knockdown GSCs generated using RCN3-specific shRNA recapitulated the results observed in cell viability and neurosphere formation studies (Appendix A). Altogether, these findings suggest that RCN3 is important for GSCs proliferation and self-renewal.

### 3.3. Identification of Global Transcriptional Changes Altered by RCN3 Knockdown

To understand the molecular mechanism underpinning RCN3 function in GSCs, we subjected the control and RCN3 knockdown GSCs to RNA-sequencing. Differentially expressed genes (DEGs) are shown (Figure 3A). Overall, 1971 genes were differentially expressed between the groups, with 1257 genes upregulated and 714 genes downregulated in RCN3 knockdown group when compared to control GSCs (Adj *p* < 0.05; |log2fold change| > 2) (Figure 3B).

To examine the biological significance of RCN3-regulated genes, we performed GSEA (gene set enrichment analysis) in GO biological process gene set collections, and the top ten upregulated or downregulated pathways are shown in Figure 3C. We found that top pathways altered by RCN3 knockdown were related to translation, ribosome, differentiation, immune response, cytokine, extracellular matrix, and stress pathways. Further, GO molecular function data also revealed that RCN3 knockdown downregulated genes were related to translation, ribosome, and cytokine signaling pathways. Upregulated genes were related to calcium ion binding, neurotransmission, and extracellular matrix (Appendix A). GSEA enrichment of GO cellular component gene sets also suggested RCN3 knockdown downregulated genes were related to ribosomes, and upregulated genes were related to the extracellular matrix and synapse (Appendix A).

Thereafter, through the protein-protein-interaction (PPI) analysis with the STRING platform and the following enrichment in the Cytoscape software, we identified the key PPI network molecules amongst the DEGs, which include a number of ribosome-related genes in the 40s ribosomal protein S (RPS) and 60s ribosomal protein L (RPL) families (Appendix A). Importantly, GSEA results demonstrated that gene sets of translation were enriched in the control group, while differentiation pathways such as neuronal differentiation, glial differentiation, and oligodendrocyte differentiation pathways were enriched in RCN3 knockdown group (Figure 3D). The expression pattern of ribosome pathway genes and differentiation pathway genes between control and RCN3 knockdown GSCs are shown in heatmaps (Figure 3E). Further in silico analysis of TIMER2 glioblastoma patient data sets also suggested that RCN3 showed a positive correlation with stemness marker CD44 and a negative correlation with differentiation markers such as ID4, MAP2, and NTRK5 (Appendix A).

Ribosomal proteins are essential factors in protein translation, and their downregulation affects protein synthesis. Given that RCN3 is a known molecular chaperone in ER lumen whose dysfunction promotes ER stress, we explored whether RCN3 attenuation regulates ribosomal and ER stress genes in GSCs. RT-qPCR results showed that the expression of several of the ribosomal constituents was downregulated, whereas ER stress markers such as BiP, XBP1, and ATF4 were upregulated in RCN3 gRNA transduced GSCs (Figure 3F–H). Further, we also observed decreased expression of stemness genes such as CD44, OCT4, and SOX9 in RCN3 gRNA transduced GSCs compared to controls (Appendix A).

To determine whether our sequencing results corroborate with tumor data sets, we explored the published CGGA data sets and obtained the mRNA expression profile of 191 glioblastoma samples from the mRNA693 dataset. With a significant threshold of adjusted *p* < 0.01 and |log2(FC)| > 1, 2870 DEGs were identified (Figure 4A,B). GO and KEGG functional enrichment of results suggested that DEGs were closely associated with multiple pathways, particularly with cytoplasmic translation, ribosome-related pathways, immune response, cytokine signaling, and extracellular matrix. (Figure 4C–E). Further GSEA analysis also suggested that the ribosome pathway is enriched in the high-RCN3 expression group compared with the low RCN3 group (Figure 4F). These results suggest that RCN3′s effects on GSCs are mediated through ribosomal pathways.

### 3.4. RCN3 Knockdown Reduced Tumor Progression in Orthotopic Xenograft Model

Given promising in vitro results demonstrating that RCN3 knockdown reduced the proliferation and self-renewal of GSCs, we next examined whether RCN3 knockdown reduced tumor progression in vivo. We orthotopically implanted patient-derived control and RCN3 shRNA-transduced GSCs; survival of tumor-bearing mice was determined. Compared to control, RCN3 knockdown significantly prolonged mice’s overall survival (Figure 5A). Tumor sections were stained using immunohistochemistry for proliferation markers Ki67, RCN3, and CD44. Results showed that Ki67 positive cells (proliferation index) and RCN3 expression were significantly lower in RCN3 knockdown tumors compared to controls (Figure 5B). We also observed that stemness marker CD44 expression is significantly reduced in RCN3 knockdown tumors compared to controls (Figure 5B). These results suggest that RCN3 knockdown reduced the glioblastoma progression in vivo.

## 4. Discussion

Reticulocalbin 3 (RCN3) encodes the Ca^2+^-binding EF-hands containing protein that functions as a molecular chaperone and is involved in protein biosynthesis and transportation in the endoplasmic reticulum. Emerging studies demonstrated that RCN3 regulates various biological processes, including extracellular matrix organization and signal transduction. However, little is known about the role of RCN3 in glioblastoma. In this study, we provided evidence that high RCN3 expression is a risk factor for poor prognosis of glioblastoma patients and plays an important role in GSCs proliferation and self-renewal. Using RNA-seq and mechanistic studies, we demonstrated that RCN3 knockdown resulted in the downregulation of translation and ribosomal pathway genes and the upregulation of stem cell differentiation-related pathway genes. Importantly, RCN3 knockdown significantly increased the survival of mice in the orthotopic glioblastoma model. Collectively, these results suggest that RCN3 plays a crucial role in glioblastoma progression.

Recent studies demonstrated that RCN3 expression in cancer is tissue specific. RCN3 is overexpressed in ESCC tissues more than in paired normal tissues and is positively associated with tumor size, lymph node metastasis, and poor outcome in patients with ESCC [23]. Further studies revealed that RCN3 is a poorer prognostic biomarker for colorectal cancer [21], overexpressed in melanoma [22], whereas it is downregulated in NSCLC [24] and osteosarcoma [25]. In our study, we observed that expression of RCN3 increased in glioblastoma tissues and was associated with a worse prognosis. It has been shown that RCN3 regulates the tumor cell characteristics as RCN3 promotes cell proliferation, invasion, and metastasis [23], and RCN3 knockdown enhances the cisplatin efficacy in ESCC. It has been shown that glioma stem cells (GSCs), which comprise a portion of glioblastoma tumors, are highly resistant to standard radiation and chemotherapy [10,33]. GSCs often contribute to tumor initiation, progression, and therapy resistance of glioblastoma [12,28,29,30,31], and their elimination is critical for the development of efficient therapeutic strategies [32]. In this study, we found that RCN3 knockdown reduces the proliferation and self-renewal ability of GSCs. Importantly, RCN3 knockdown increased the survival of tumor-bearing mice in orthotopic models. Further, supporting the role of RCN3 in the self-renewal of GSCs, our sequencing studies showed that RCN3 knockdown increased the expression of several genes involved in stem cell differentiation pathways.

In addition to its role in the secretory pathway, RCN3 is also involved in other biological pathways. For example, a proteomic study reveals that reticulocalbin 3 is transiently associated with the precursor of subtilisin-like proprotein convertase (proPACE4) and plays an essential role in the biosynthesis of PACE4 [17]. In human cardiac fibroblasts, RCN3 exerts a negative effect on collagen production, as treatment with recombinant RCN3 decreased collagen expression through Akt phosphorylation [16]. Further, RCN3 deficiency significantly blunted LPS-induced acute lung injury and alveolar inflammation via reduced activations of NF-κB signaling and NLRP3-dependent inflammasome [34]. RCN3 is recognized as a regulator in pulmonary surfactant homeostasis and is largely studied on its roles of proper biosynthesis and transport of pulmonary surfactant-associated protein [20,34,35,36]. Although RCN3 is an ER-residing protein, interestingly, a recent study also discussed the possible cytoplasmic translocation of RCN3, which might regulate the activation process of NF-κB [34]. RCN3 promoted the expression of MMP-2 and MMP-9 by regulating the inositol 1,4,5-trisphosphate receptor 1 (IP3R1)–Ca^2+^–calcium/calmodulin-dependent protein kinase II–c-Jun signaling pathway [23]. RCN3-deleted mice also displayed enhanced alveolar epithelial cell (AEC) apoptosis and ER stress after bleomycin treatment, which was confirmed by in vitro studies both in primary AECIIs and mouse lung epithelial cells [35]. In agreement with these studies, our results showed that RCN3-regulated genes are involved in ER stress pathways. Interestingly, our study results also demonstrated that translation and ribosomal pathways are the top downregulated pathways in RCN3 knockdown GSCs. We speculate that both ER location-dependent and independent functions of RCN3 may play a role in regulating these pathways. However, future studies are clearly needed to understand the in-depth mechanisms of RCN3 regulation of ribosomal, immune, and cell differentiation pathways.

Protein synthesis is a tightly regulated and coordinated process involving the action of ribosomes and a set of translation factors. Ribosomes are composed of ribosomal RNAs (rRNAs) and ribosomal proteins (RPs). Several recent studies demonstrated that RPs are implicated in a variety of pathological processes, including tumorigenesis and cell transformation [37]. Deregulation of RPs impacts ribosomal biogenesis, which can impair cell survival, growth, and proliferation. Although ribosomal proteins are known for playing an essential role in ribosome assembly and protein translation, their ribosome-independent functions, such as tumorigenesis, immune signaling, and development, have also been greatly appreciated [38]. Interestingly, an in vivo genome-wide CRISPR activation screen in breast cancer patient-derived CTCs identified that genes coding for ribosomal proteins and regulators of translation were enriched in this screen [39]. In our study, we observed that protein translation and ribosomal pathways were decreased in RCN3 knockdown cells, suggesting that RCN3 knockdown-mediated effects on GSCs may partly be attributed to decreased expression of genes involved in protein translation.

In summary, we showed that RCN3 is upregulated in glioblastoma, and high RCN3 expression was correlated with clinical progression and worse prognosis. Moreover, the knockdown of RCN3 inhibited GSCs progression in vitro and in vivo, suppressed ribosome pathways, and activated stem cell differentiation pathways. RCN3 expression may be an important diagnostic and prognostic factor, and glioblastoma patients with high RCN3 expression may exhibit poor survival rates compared to low RCN3 expression group; thus, RCN3 will serve as a valuable therapeutic target for patients with glioblastoma.

## 5. Conclusions

This study provides evidence that RCN3 is upregulated in glioblastoma and associated with poor overall survival. Knockdown or knockout of RCN3 decreased the proliferation and self-renewal ability of GSCs. Mechanistic studies revealed that RCN3 knockdown resulted in the downregulation of translation and ribosomal pathways and the upregulation of stem cell differentiation pathways. Using mouse orthotopic glioblastoma models, we showed that the knockdown of RCN3 resulted in enhanced survival of tumor-bearing mice. Collectively, these results suggest that RCN3 is a novel mediator of glioblastoma progression.

## Figures and Tables

**Figure 1 cancers-15-02008-f001:**
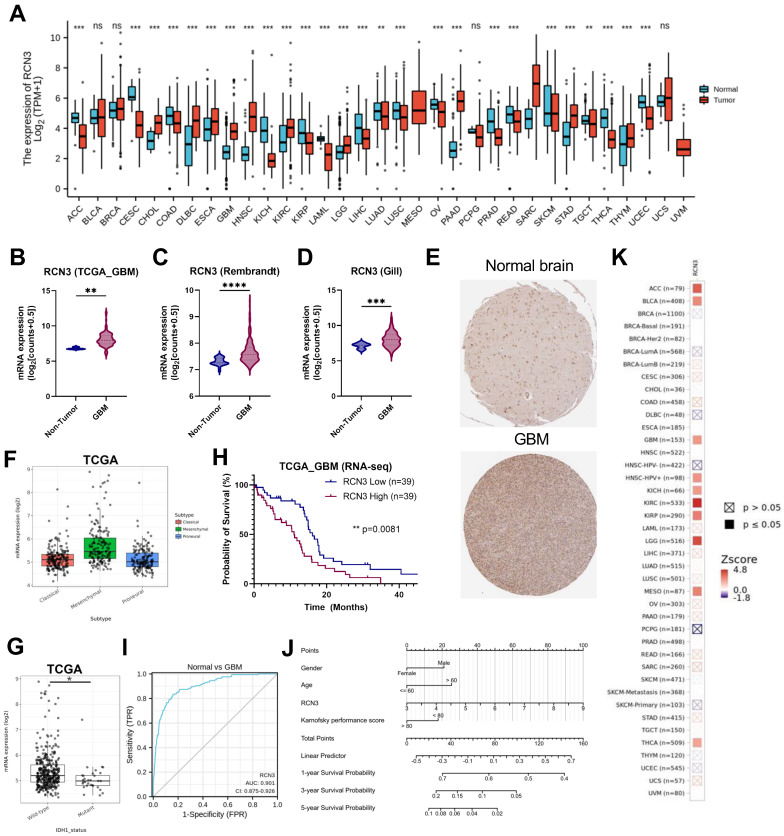
RCN3 is overexpressed in glioblastoma (GBM). (**A**) Expression of RCN3 mRNA in different types of human cancers in the UCSC XENA database; (**B**–**D**) RCN3 mRNA expression in normal tissue and glioblastoma from various data sets of TCGA, REMBRANDT, and gill, respectively. (**E**) RCN3 protein expression in normal tissue and glioblastoma from the Human Protein Atlas database. (**F**) RCN3 expression in different molecular subtypes of TCGA glioblastoma data set. (**G**) RCN3 expression in IDH wild-type and IDH mutant tumors in TCGA data set. (**H**) Kaplan–Meier curve of overall survival (OS) in TCGA glioblastoma data sets. (**I**) ROC curve analysis shows the diagnostic value of RCN3 in TCGA glioblastoma data set. (**J**) Nomogram integrating the clinicopathological variables and RCN3 expression level to predict the 1-, 3-, and 5-year survival probabilities of glioblastoma patients. The points for each attribute are calculated according to the corresponding status and the matching location in the first row of points. The sum of points is then used to predict the survival possibility for certain patients. (**K**) Prognosis correlation analysis of RCN3 in glioblastoma using TIMER2.0 database. ns, *p* ≥ 0.05; *, *p* < 0.05; **, *p* < 0.01; ***, *p* < 0.001, ****, *p* < 0.0001 by Student’s *t*-test.

**Figure 2 cancers-15-02008-f002:**
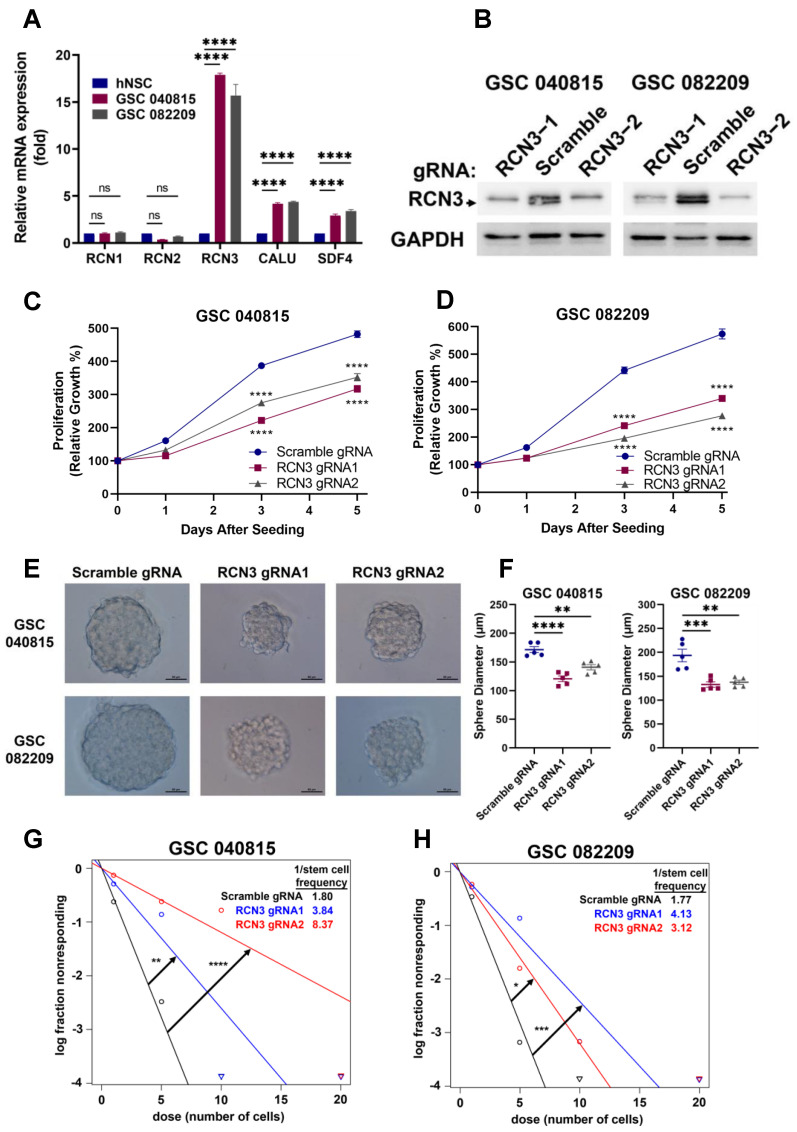
RCN3 knockout reduces the proliferation and self-renewal of GSCs. (**A**) Expression of CREC family members was determined in hNSCs and GSCs using RT-qPCR assays (n = 3 replicates). (**B**) GSC 040815 and GSC 082209 cells were transduced with either scramble or RCN3-specific gRNA1 or gRNA2, and RCN3 levels were confirmed using Western blotting. (**C**,**D**) Cell proliferation rates of scramble or RCN3 gRNA-transduced GSCs were determined using CellTiter-Glo assay (n = 3). (**E**,**F**) Neurosphere formation (n = 5 representative spheres per group) or (**G**,**H**) self-renewal (n = 24 wells per group) of scramble and RCN3-gRNA transduced GSCs was determined using sphere formation or limiting dilution assay, respectively. Spheres were sized and analyzed using ImageJ. ELDA plot of number of cells seeded (dose) versus log fraction non-responding was created using ELDA software, circles indicate data points, and triangles show data points that are not visible within the plot. Self-renewal was quantified as the reciprocal of stem cell frequency (1/Stem cell freq) determined by ELDA software. Higher self-renewal ability corresponds to lower 1/Stem cell freq ratio. Scale bar of 50 μm. * *p* < 0.05, ** *p* < 0.01, *** *p* < 0.001, **** *p* < 0.0001, by *t*-test or one-way ANOVA.

**Figure 3 cancers-15-02008-f003:**
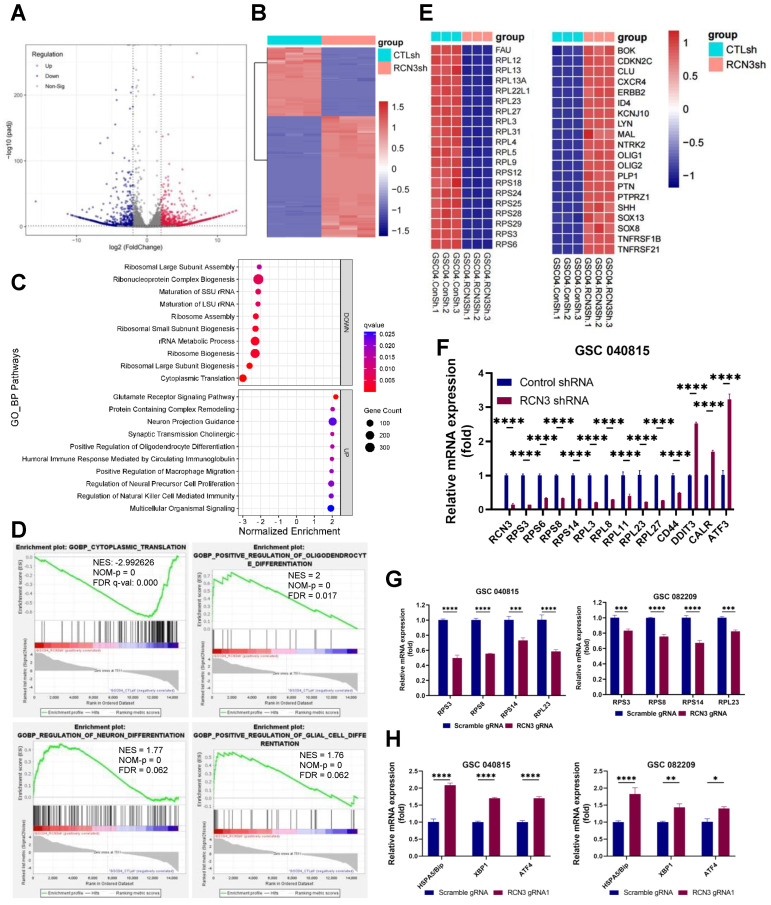
RCN3 knockdown altered the expression of genes related to translation, ribosome, and stem cell differentiation. GSC040815 cells were transduced with either control or RCN3 shRNA and subjected to RNA sequencing. (**A**) Volcano plots showing the differentially expressed genes (DEGs) between the control and RCN3 knockdown GSCs. (**B**) Heatmap displaying the expression pattern of DEGs between groups. (**C**) Top enriched GO terms of biological processes from GSEA analysis are shown in bubble plots. (**D**) GSEA plots showing enrichment of RCN3-regulated genes in signatures of cytoplasmic translation, regulation of neuron differentiation, glial cell differentiation, and oligodendrocyte differentiation gene sets. (**E**) Heatmaps showing top 20 downregulated genes in ribosomal (**left**) and stem cell differentiation (**right**) genes. (**F**–**H**), RT-qPCR validation of downregulated ribosomal or upregulated ER stress genes in GSCs transduced with RCN3 shRNA (**F**) or RCN3 gRNA (**G**,**H**), with n = 3 per assay. * *p* < 0.05, ** *p* < 0.01, *** *p* < 0.001, **** *p* < 0.0001, by Student’s *t*-test.

**Figure 4 cancers-15-02008-f004:**
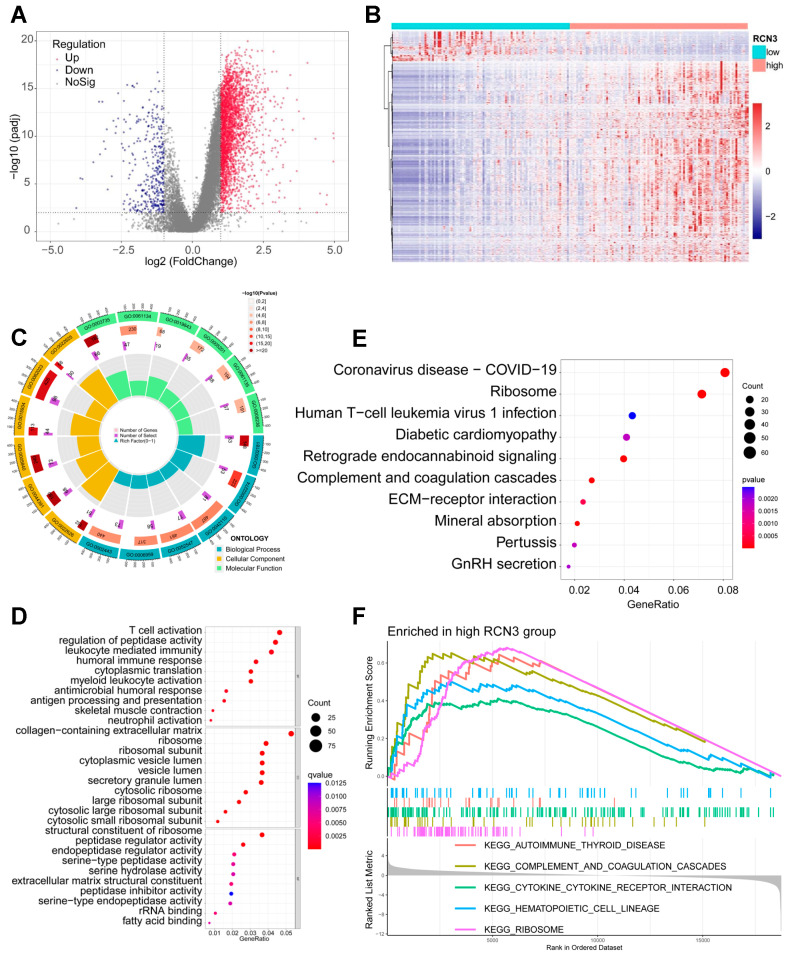
In silico analysis of CGGA data set suggested RCN3 expression is correlated with the expression of ribosome pathway genes. (**A**) The volcano plot shows the differentially expressed genes between low and high RCN3 expression glioblastoma samples. (**B**) Heatmap showing the DEGs expression pattern between low and high RCN3 expression glioblastoma samples. (**C**–**E**) GO (Gene Ontology) and KEGG (Kyoto encyclopedia of genes and genomes) functional enrichment of DEGs. (**F**) GSEA enrichment analysis of KEGG gene sets to compare the gene expression differences between the low and high RCN3 glioblastoma samples. Each line represents one particular gene set with unique color, and upregulated genes are located on the left approaching the origin of the coordinates; by contrast, the downregulated ones lay on the right of *x*-axis. Only gene sets with NOM *p* < 0.05 and FDR q < 0.25 were considered significant. Only several leading gene sets were displayed in the plot.

**Figure 5 cancers-15-02008-f005:**
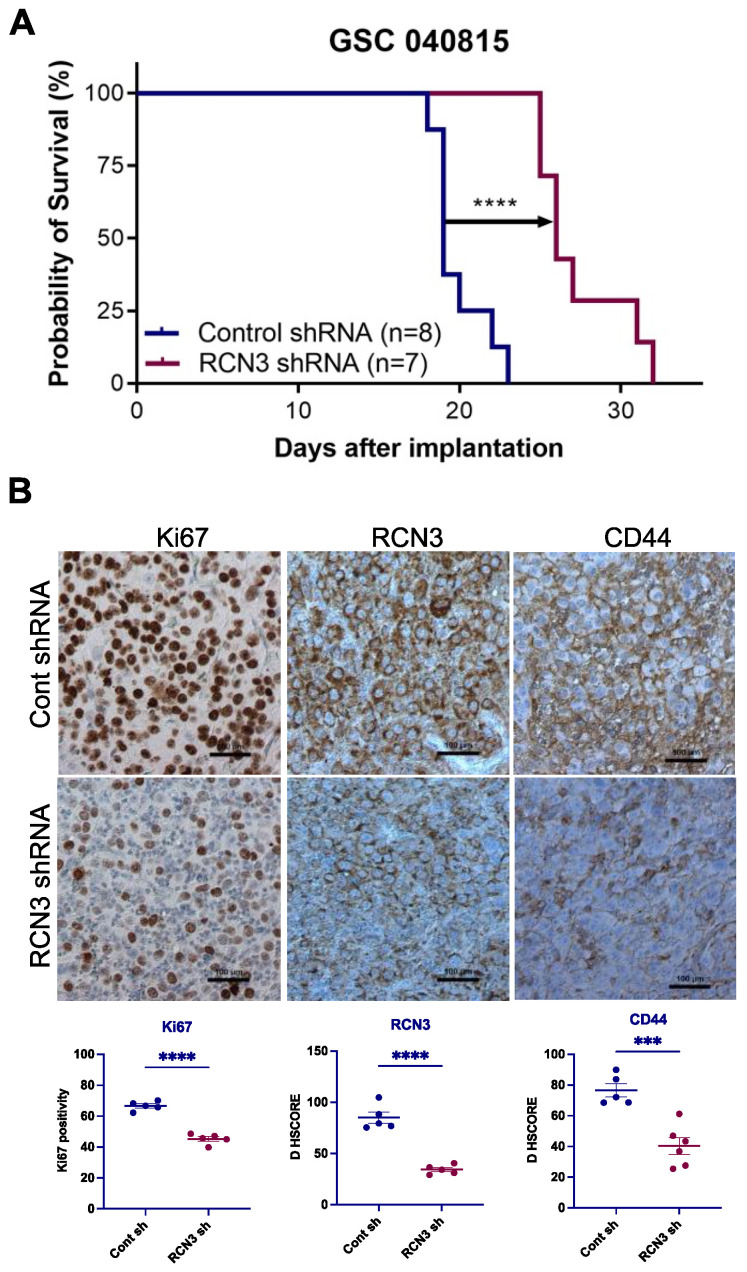
RCN3 knockdown enhances the survival of tumor-bearing mice. (**A**) GSC 040815 cells that stably express control (n = 8) or RCN3 (n = 7) shRNA were implanted intracranially into NOD-SCID mice, and survival was plotted using Kaplan–Meier curves. **** *p* < 0.0001 by log-rank test. (**B**) Tumor tissues collected from control and RCN3 knockdown groups were subjected to IHC staining for Ki67 (nuclear), RCN3 (cytoplasmic), and CD44 (membrane), and staining intensity was quantitated. *** *p* < 0.001, **** *p* < 0.0001 by student’s *t*-test.

## Data Availability

The RNA-seq data were deposited in the GEO database under a GEO accession number GSE226990.

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
