# Peer review of "Reticulocalbin 3 Is a Novel Mediator of Glioblastoma Progression"

_cancers, 2023, doi:10.3390/cancers15072008_

Round 1

Reviewer 1 Report

The authors have convincingly shown that RCN3 RNA is overexpressed in glioblastoma and is a reliable bad prognosis marker. They provide evidence that RCN3 knockdown reduces the viability and self-renewal of glioma stem cells (GSC), correlated with large scale transcriptomic alterations characterized by down regulation of genes involved in translation and ribosome formation and upregulation of genes involved in stem cell differentiation. Importantly, they demonstrate that RCN3 knockdown increases the survival of mice implanted with patient-derived GSCs. Overall these results suggest that RCN3 might be an interesting therapeutic target.

The paper presents interesting and well performed experiments. The figure legends would benefit from being more detailed and  the authors should spell out acronyms, whose meanings might not be obvious to many readers.

Minor points:

-          Line 211 “ROC curve”: give the full name.

-          Line 212: “AUC”: give the full name.

-          Line 215: “Fig. 1G”: should be 1I ?

-          Line 216: “Fig. 1H” should be 1J ?

-          Figure 1 legend: (A): The data shown are probably RNA levels: state this clearly. (B-D): give more experimental details.

-          The resolution of Figure 1F is poor.

-          Line 247: “Cell-titer Glo assays”: give the full name of “Glo”.

-          Figure 2: (E-F) the insets with “1/Stem Cell Freq RCN3 k/d” are difficult to understand: this is a good example of general lack of sufficient information. Overall, more experimental details should be given for panels E-F.

-          Figure 2H: what is the upper band in the RCN3 western? The authors do not comment. This band was not seen in the western of Figure 2A.

-          Line 270: “GSEA”: give full name.

-          Figure 3 legend: (D-F): should be (C-E) ? “(G) Key network…” should be (F) ?

-          Figure 3, panel F: more explanations needed.

-          Line 315: “downreguatled” should be “downregulated”.

-          Figure 5: Letters “A” and “B” are missing for the top panels. Overall, panels A, B and C convey little information to the readers as presented.

-          Figure 5F: more explanations needed.

-          Line 328: “KEGG”: give full name.

-          Figure 6B: the authors should comment on the subcellular localization of RCN3 staining.

-          Line 374-375: are highly resistant (not “highly resistance”).

-          Line 383: RCN3 is also involved… (“is” missing).

-          Line 401: regulated genes are involved…(“are” missing)

-          Line 403: pathways (“s” missing).

Author Response

Reviewer 1

Comment: The paper presents interesting and well performed experiments. The figure legends would benefit from being more detailed and the authors should spell out acronyms, whose meanings might not be obvious to many readers.

Response: Thank you very much for your positive review and valuable suggestions. We have now provided elaborated figure legends and spelled-out acronyms.

Minor points:

Comment: Line 211 “ROC curve”: give the full name;  Line 212: “AUC”: give the full name;  Line 215: “Fig. 1G”: should be 1I ?;  Line 216: “Fig. 1H” should be 1J ?

Response: We have corrected these minor points.

Comment:  Figure 1 legend: (A): The data shown are probably RNA levels: state this clearly. (B-D): give more experimental details;  The resolution of Figure 1F is poor; Line 247: “Cell-titer Glo assays”: give the full name of “Glo”. Figure 2: (E-F) the insets with “1/Stem Cell Freq RCN3 k/d” are difficult to understand: this is a good example of general lack of sufficient information. Overall, more experimental details should be given for panels E-F.  

Response: We have corrected these minor points.

Comment:  Figure 2H: what is the upper band in the RCN3 western? The authors do not comment. This band was not seen in the western of Figure 2A.

Response: We thank the reviewer for pointing out this. We believe that the upper band could be a non-specific band, which is also shown in the corresponding manufacturer data sheet. Since we run the samples detailed in Figure 2A on gradient SDS gel, this nonspecific band was separated out, hence we did not see this close to the original band, and it was cropped out. For clarification, we have rerun this sample and presented the uncropped versions of all RCN3 western blots in supplementary figure 4.

Comment:  Line 270: “GSEA”: give full name; Figure 3 legend: (D-F): should be (C-E) ? “(G) Key network…” should be (F) ?; Figure 3, panel F: more explanations needed; Line 315: “downreguatled” should be “downregulated”; Figure 5: Letters “A” and “B” are missing for the top panels. Overall, panels A, B and C convey little information to the readers as presented.

Response: These minor errors have been fixed. For Figure 5A-B, we try to show the general expression landscape difference between the low- and high- RCN3 expression GBM samples from the CCGA data set. And Figure 5C summarized the statistical characteristics of the top GO enrichment pathways of DEGs, and the details of those pathways are exhibited in Figure 5D.

Comment:  Figure 5F: more explanations needed; Line 328: “KEGG”: give full name; Figure 6B: the authors should comment on the subcellular localization of RCN3 staining; Line 374-375: are highly resistant (not “highly resistance”); Line 383: RCN3 is also involved… (“is” missing); Line 401: regulated genes are involved…(“are” missing); Line 403: pathways (“s” missing).

Response: We have corrected these minor points.

Reviewer 2 Report

In this manuscript of Yi He, the authors have investigated the functional role of Reticulocalbin 3 (RCN3 ) in glioma. This is an interesting and potentially important work on the role RCN3.as potential therapeutic target in glioma patients. The experiments are well executed, the results are logically presented, and the discussion is thorough.

Comments and suggestion for authors

The authors did not specify why the U251 glioma cell line they used in their study. There are many other glioma cells lines, which are available from the ATCC. What was the rational of using this particular cell line? Have you tried to use other glioma cell lines?

How glioma patients can “profit” from the results of your study? What patients with elevated levels of RCN3 (=poor prognosis) could do? To my mind the manuscript would benefit from discussion of this issue (please add a couple of sentences).

In vivo, in vitro must be written in italics

Author Response

Reviewer 2

Comment: The authors did not specify why the U251 glioma cell line they used in their study. There are many other glioma cells lines, which are available from the ATCC. What was the rationale of using this particular cell line? Have you tried to use other glioma cell lines?

Response:

We appreciate your thorough analysis and helpful recommendations. We used two different patient-derived glioma stem cells (GSC040815 and GSC082209) as model cells in this study, and the characterization of these GSCs was reported in our recent publication in “Neuro-Oncology” (PMID 36652263). Several studies affirm that patient-derived GSCs have better clinical relevance to glioblastoma as they retain the genetics, histological, and molecular profile similar to the primary tumors, unlike the serum-grown ‘classic’ cell lines such as U87 and U251, which may not truly resemble human GBM features (PMID 31519690, 16697959, 26629530). Considering the reviewer concern, we have removed the U251 data and conducted most of the experiments using two different primary GSCs lines.

Comment: How glioma patients can “profit” from the results of your study? What patients with elevated levels of RCN3 (=poor prognosis) could do? To my mind the manuscript would benefit from discussion of this issue (please add a couple of sentences).

Response: We thank the reviewer for this suggestion. We have now added information about the relevance of RCN3 high expression and GBM prognosis in the discussion section.

Comment: In vivo, in vitro must be written in italics.

Response:  These are corrected.

Reviewer 3 Report

The authors describe a new function of the RCN3 gene in glioblastoma. To do this, He et al did in vivo and in vitro experiments, decreasing the expression of RCN3 and describing an oncogenic role of this gene in glioblastoma. In addition, they validate their results in two of the most important databases in glioblastoma: TCGA and CGGA. The work is well done, however there are certain aspects that could be improved.

Major points:

The authors refer to glioblastoma as GBM, I recommend that they refer to GB/glioblastoma; or GBM/glioblastoma multiforme. On the other hand, there is no mention in the entire article about the status of IDH. If the lines have been isolated from IDH wt patients, special emphasis should be placed on this and introduction and discussion should be reformulated accordingly. I would recommend to follow the 2021 WHO Classification of Tumors of the Central Nervous System and refer glioblastoma as a GB IDH wt.

Is it possible to measure the expression of RCN3 and the rest of family members (RCN1, RCN2, SDF4, and CALU) by RT-PCR and WB in the 3 cell lines: U251, GSC0408115 and GSC082209? Would it be possible to include non-tumor cell lines such as astrocytes or microglia? Would it be possible to measure RCN3 levels in tumor tissue and in healthy tissue/low grade gliomas? I recommend to correlate the expression of RCN3 with GSCs markers (Sox2, CD133, CD44, Nestin).

Have the lines GSC0408115 and GSC082209 been previously published? If not, a functional and molecular characterization should be included. Measure by mRNA and WB the expression of tumor and/or stem cell markers such as SOX2, SOX9, CD133, CD44, Nestin; proliferation. Also the differentiation potential to neuron, astrocyte and oligodendrocyte.

Which specific mutations do patients and lines present? For example; IDH, EGFR vIII, TERT, ATRX, p53, Ki67index, MGMT methylation. If this information is not available, it could be determined in the cell lines themselves. Also, clinical information such as survival, age, gender should be included.

In the functional assays section, I don't understand why some assays are done using shRNAs and others using CRISPR/Cas9. Above all, I am very surprised that when the authors do CRISPR Cas9 you do not get a RCN3 knockout and that the highest levels of inhibition are achieved by doing shRNA. What is the explication of that? I recommend to perform one of the two techniques (shRNA or CRISPR) in the 3 cell lines (at least in GSC0408115 and GSC082209) using two shRNAs or 2 different Cas9 clones. In the same way, a validation of the CRISPR/Cas9 technique and a confirmation that the other members of the RCN3 family are not affected must be presented. I recommend additionally performing a differentiation study on cells with shRNAs or Cas9 depleted. Also, I recommend to measured GSCs markers in the RCN3 knockdown cells.

It would be interesting to present a validation of several RNA-Seq pathways in the lines GSC0408115 and GSC082209. I recommend to merge figures 3 and 4.

In Figure 5, I recommend going directly to a validation of the altered pathways observed in the RNA-Seq in the CGGA cohort. Skip DESeq analysis.

In vivo experiment should include additional glioblastoma and glioma stem cell markes such as Sox2, Sox9, CD133, etc.

Minor points:

Scale bars are missing.

Experimental n is missing in all figures.

Experimental protocol of lentiviral production is missing.

Authors should include all primers for RT-qPCR and the sequence of shRNAs and gRNAs. Also, authors should include the concentrations of 1ry and 2ry antiboides used for wb.

I would recommend to perform primary and secondary neurosphere formation assays. If not I don’t agree with the sentence in lines 246-247 “RCN3 knockdown significantly reduced the neurosphere self-renewal”

RNA extraction is performed with a treatment with DNAase?

Authors should increase the quality of some images (1F, 3E,F)

In which dataset ROC curves are performed?

Line 215 Fig 1I instead of Fig 1G is not correct. Authors should explain the implication of 1m 3 and 5 years survival prediction.

Figure 3: please have a logical figure order.

Author Response

Reviewer 3

Comment: The authors refer to glioblastoma as GBM, I recommend that they refer to GB/glioblastoma; or GBM/glioblastoma multiforme. On the other hand, there is no mention in the entire article about the status of IDH. If the lines have been isolated from IDH wt patients, special emphasis should be placed on this and introduction and discussion should be reformulated accordingly. I would recommend to follow the 2021 WHO Classification of Tumors of the Central Nervous System and refer glioblastoma as a GB IDH wt.

Response: We thank the reviewer for this excellent suggestion. To avoid the confusion of terminology of glioblastoma vs. glioblastoma multiforme and GB vs. GBM, we have used the term “glioblastoma” throughout the manuscript. Further, as per the 2021 WHO CNS5 classification, we referred to glioblastoma as glioblastoma IDH-wild type in the introduction section.

Our GSCs are established from IDH wild-type glioblastoma patients. Further exploration of TCGA and CGGA data sets also showed significantly higher RCN3 expression in IDH wild-type tumors compared to IDH mutant tumors (Fig. 1G; Supplementary Fig. 1B).

Comment: Is it possible to measure the expression of RCN3 and the rest of family members (RCN1, RCN2, SDF4, and CALU) by RT-PCR and WB in the 3 cell lines: U251, GSC0408115 and GSC082209? Would it be possible to include non-tumor cell lines such as astrocytes or microglia? Would it be possible to measure RCN3 levels in tumor tissue and in healthy tissue/low grade gliomas? I recommend to correlate the expression of RCN3 with GSCs markers (Sox2, CD133, CD44, Nestin).

Response: We thank the reviewer for this excellent suggestion. As per the reviewer's suggestion, we have measured the CREC family members RCN1, RCN2, SDF4, and CALU by RT-PCR in GSCs as well as human normal neural stem cells (hNSCs). Our result showed that RCN3 is highly expressed in GSCs compared to hNSCs. Unfortunately, we do not have glioblastoma tissue to examine the RCN3 expression immunohistochemically. However, we observed high RCN3 expression in glioblastoma tissue and this data was extracted from the human protein atlas database. Further, we did the correlation analysis of RCN3 with CD44, sox2, cd133, and nestin. We observed a significant positive correlation with CD44, however, no significant trend was observed with other markers.

Comment: Have the lines GSC0408115 and GSC082209 been previously published? If not, a functional and molecular characterization should be included. Measure by mRNA and WB the expression of tumor and/or stem cell markers such as SOX2, SOX9, CD133, CD44, Nestin; proliferation. Also the differentiation potential to neuron, astrocyte and oligodendrocyte.

Response: Thanks for the suggestion. Yes, the characterization of these cell lines was published in our recent publication in “Neuro-Oncology” (Alejo et al., Neuro-Oncology, 2023; PMID 36652263). These cell lines were well characterized, RNA-seq verified, and subtyped as per TCGA classification.

Comment: Which specific mutations do patients and lines present? For example; IDH, EGFR vIII, TERT, ATRX, p53, Ki67index, MGMT methylation. If this information is not available, it could be determined in the cell lines themselves. Also, clinical information such as survival, age, gender should be included.

Response: Thanks for this insightful suggestion. GSC040815 and GSC082209 were MGMT methylated, and these cell lines were quite sensitive to temozolomide. Our recent publication reported these findings (Alejo et al., Neuro-Oncology, 2023; PMID 36652263).

Unfortunately, we don’t have information regarding the EGFR vIII, TERT, ATRX, p53, Ki67index of the GSCs at this time, and getting IRB approval to access the patient's data is time taking process. However, we value the reviewer's comment, and we will explore the possible association between RCN3 and the clinical information of patients in our future studies.

Comment: In the functional assays section, I don't understand why some assays are done using shRNAs and others using CRISPR/Cas9. Above all, I am very surprised that when the authors do CRISPR Cas9 you do not get a RCN3 knockout and that the highest levels of inhibition are achieved by doing shRNA. What is the explication of that? I recommend to perform one of the two techniques (shRNA or CRISPR) in the 3 cell lines (at least in GSC0408115 and GSC082209) using two shRNAs or 2 different Cas9 clones. In the same way, a validation of the CRISPR/Cas9 technique and a confirmation that the other members of the RCN3 family are not affected must be presented. I recommend additionally performing a differentiation study on cells with shRNAs or Cas9 depleted. Also, I recommend to measured GSCs markers in the RCN3 knockdown cells.

Response: We are appreciative of the reviewer's criticism. We have used shRNA for the initial characterization of RCN3 effect on GSCs functional assays. Once we got the evidence that RCN3 knockdown reduced the GSCs proliferation, we then went on to generate CRISPIR-Cas9 cells to further validate RCN3 effect on GSCs. Regarding RCN3-gRNA transduced cells, we used pooled RCN3 gRNA transduced (not the individual clones) cells to confirm the knockout and any residual band in the western blot could have resulted from non-transduced cells or single copy knockdown cells.

As per the reviewer's suggestion, we have used two different RCN3 gRNAs in two different GSCs (GSC040815 and GSC082209) and performed the functional assays. To confirm the specific knockdown of RCN3 but not other members, we have looked at the expression of RCN3 and other members in GSCs. RT-qPCR assay results showed that RCN3 gRNA transduction did not have major changes in the expression of other CREC family members (Supplementary Fig. 2F). Further, a closer examination of our RNA-seq expression data also showed no downregulation of other CREC family members in RCN3 knockdown cells.

As per the reviewer's suggestion, we have checked the expression of stemness markers CD44, sox2, sox9, and oct4 and found that the expression of these markers was significantly reduced in RCN3 gRNA-transduced cells compared to scramble controls (Supplementary Fig. 3E).  We are unable to perform differentiation assay at this time due to time limitation for the revision, and we will explore this in our future studies.

Comment: It would be interesting to present a validation of several RNA-Seq pathways in the lines GSC0408115 and GSC082209. I recommend to merge figures 3 and 4.

Response: We validated the genes identified in RNA-seq pathways using RT-qPCR (Fig. 3F,G,H). We also merged the figures 3 and 4.

Comment: In Figure 5, I recommend going directly to a validation of the altered pathways observed in the RNA-Seq in the CGGA cohort. Skip DESeq analysis.

Response: Thanks for the suggestion. In Figure 5A-B, we try to show the general expression landscape difference between the low- and high- RCN3 expression glioblastoma samples from the CGGA data set. And Figure 5C summarized the statistical characteristics of top GO enrichment pathways of DEGs, and the details of those pathways are exhibited in Figure 5D.

Comment: In vivo experiment should include additional glioblastoma and glioma stem cell markers such as Sox2, Sox9, CD133, etc.

Response: Since we observed a strong correlation between RCN3 and CD44 expression, we performed IHC on tissue sections for GSCs marker CD44. Our results showed that CD44 expression is significantly downregulated in RCN3 knockdown tumors compared to controls.

Minor points:

Comment: Scale bars are missing; Experimental n is missing in all figures; Experimental protocol of lentiviral production is missing.

Response: We have provided this information.

Comment: Authors should include all primers for RT-qPCR and the sequence of shRNAs and gRNAs. Also, authors should include the concentrations of 1ry and 2ry antibodies used for wb.

Response: We have now included the primer sequences, shRNA, gRNA information, and antibody dilutions.

Comment: I would recommend to perform primary and secondary neurosphere formation assays. If not I don’t agree with the sentence in lines 246-247 “RCN3 knockdown significantly reduced the neurosphere self-renewal”

Response: Our primary cell lines were cultured as neurospheres in stem cell permissive media and have never been exposed to serum. As a result, they have maintained their stemness and we believe that secondary sphere formation experiments would not accurately measure their self-renewal ability. In fact, the standard method in the field for evaluating self-renewal in GSCs, as we use in this manuscript, has been the use of the extreme limiting dilution assay (PMID: 23995067, 34395790, 36652263, 26859459).

Comment: RNA extraction is performed with a treatment with DNAase ?

Response: Yes, DNAase treatment was included in RNA isolation steps (Qiagen RNEasy Kit Cat# 74104).

Comment: Authors should increase the quality of some images (1F, 3E,F); In which dataset ROC curves are performed? Line 215 Fig 1I instead of Fig 1G is not correct. Authors should explain the implication of 1m 3 and 5 years survival prediction; Figure 3: please have a logical figure order.

Response: We have corrected all these minor suggestions. The ROC curve analysis was performed with TCGA data. Related information has been added to the figure legend. In nomoplot, the points for each attribute are calculated according to the corresponding status and the matching location in the first row of points. The sum of points is then used to predict the 1-, 3-, and 5-year survival possibility for certain patients. More detailed explanation has been added to the corresponding figure legend.

Round 2

Reviewer 3 Report

The authors have made all the changes I suggested. However, my major concern is methodological.

The authors don’t explain why doing CRISPR-Cas9 they get a kd and not a ko. Authors should explain this in detail and provide sanger sequencing confirmation of the CRISPR-Cas9.  In my understanding CRISPR-Cas9 completely and permanently silences the gene at the DNA level.

Minor points:

Authors should precise that hNSC are iPSC derived from neonatal foreskin fibroblasts.

Author Response

We thank the reviewer again for critically reading our manuscript and for helpful suggestions.

Comment: The authors don’t explain why doing CRISPR-Cas9 they get a kd and not a ko. Authors should explain this in detail and provide sanger sequencing confirmation of the CRISPR-Cas9.  In my understanding CRISPR-Cas9 completely and permanently silences the gene at the DNA level.

Response: In theory, CRISPR-Cas9 knockout should result in complete gene knockout in the cells, and we concur with the reviewer on this point. The Cas9-gRNA complex cuts both the DNA strands, and the DNA break is then repaired by non-homologous end joining, a pathway that is prone to error and can introduce insertion or deletion mutations that can cause frameshifts and a premature termination codon (PTC) in the expressed transcript, leading to the decay of mRNA and the degradation of aberrant peptide products. Because of the nature of the repair used by CRISPR system, pooled clones exhibit a mosaic of knockdown due to diverse repair events that may occur. As such, residual expression of the targeted gene is frequently seen in some clones, especially when pooled clones are created.   

The causes of residual expression were recently validated in a publication in “Nature Methods” (Nature Methods, 2019, issue 16, pages 1087–1093; PMID: 31659326). They discovered two causal mechanisms—translation re-initiation leading to target protein isoforms with internal sequence deletions or skipping of the edited exon leading to protein isoforms with residual protein expression—for about one-third of the 193 CRISPR targets. The scope of our study, however, does not include systemic characterization of the clones harboring residual expression of RCN3. In our study, we did observe faint expression of RCN3 in one GSC model (RCN3-gRNA1 of GSC082209) and this residual band in the western blot could have resulted from non-transduced cells or single copy knockdown cells. However, this pool of cells exhibited the expected phenotype of reduction of RCN3 as well as reduction of cell proliferation and self-renewal and are similar to “complete” RCN3 knockout cells. Further, the phenotype is confirmed by two independent gRNAs in two different GSCs models, whose levels were validated using western blotting as described in the methods, and we believe that the phenotypic effect of RCN3 reduction is convincing. We have now added additional information in the methods sections. 

Information added in the Methods section: 

The knockout of RCN3 in gRNA transduced GSCs was confirmed by Western blotting. To prevent the artifact of choosing a single clone, we have employed pooled clones. We detected faint expression of RCN3 in GSC082209 RCN3-gRNA1 cells, and this expression could have resulted from non-transduced cells or single copy knockdown cells and the intricate nature of CRISPR knockout mediated DNA damage repair.

Minor points:

Authors should precise that hNSC are iPSC derived from neonatal foreskin fibroblasts.

Response: We have included this information in the methods section.

Round 3

Reviewer 3 Report

Authors have made all changes I suggested.